# Baijiu Vinasse Extract Scavenges Glyoxal and Inhibits the Formation of *N*^ε^-Carboxymethyllysine in Dairy Food

**DOI:** 10.3390/molecules24081526

**Published:** 2019-04-18

**Authors:** Yuzhen Wang, Huilin Liu, Dianwei Zhang, Jingmin Liu, Jing Wang, Shuo Wang, Baoguo Sun

**Affiliations:** 1Beijing Advanced Innovation Center for Food Nutrition and Human Health, Beijing Engineering and Technology Research Center of Food Additives, Beijing Technology and Business University, 11 Fucheng Road, Beijing 100048, China; wyzdzg999@163.com (Y.W.); zhangdianwei1205@163.com (D.Z.); liujingmin@nankai.edu.cn (J.L.); wangshuo@nankai.com (S.W.); sunbg@th.btbu.edu.cn (B.S.); 2Tianjin Key Laboratory of Food Science and Health, School of Medicine, Nankai University, Tianjin 300071, China

**Keywords:** *N*^ε^-carboxymethyllysine, phenolic acids, glyoxal, vinasse, inhibition, dairy

## Abstract

The inhibitory effects of baijiu vinasse extract and its phenolic acid compounds on the *N*^ε^-carboxymethyllysine (CML) formation from dairy food were investigated. The inhibitory effect of the baijiu vinasse extract against CML formation was 43.2% in the casein and D-glucose model, which used 6 mL of the 70% acetone extract at 60 °C for 40 min. The HPLC-MS/MS profiles of the vinasse extract indicated that vanillic, chlorogenic, *p*-coumaric, sinapic, caffeic, ferulic, and syringic acids were seven major phenolic acid compounds. Furthermore, the inhibitory mechanism of the phenolic acid compounds in the model of dairy food was discussed by the trapping and scavenging of glyoxal. The results of this study exhibit that seven major antioxidant phenolic acid compounds may play important roles in the antioxidant activity and CML inhibition of the vinasse extract in a model of dairy foods.

## 1. Introduction

The thermal processing of food, which is mainly attributed to the Maillard reaction, can preserve flavor and color and enrich nutrition. The Maillard reaction, also known as the nonenzymatic browning reaction, is a complex reaction between carbonyl groups in reducing sugar and free amino groups in amino acids, peptides, or proteins to form Maillard reaction products (MRPs) [1]. *N*^ε^-carboxymethyllysine (CML) is an atypical Maillard reaction product from biological systems and food matrices [2]. Dietary CML and other MRPs are important contributors to degenerative diseases [3], and Alzheimer’s disease [4], because the gastrointestinal tract is its absorption location and may lead to an increase in the level of CML and other MRPs within the body [5]. Dietary CML is also a risk factor for inflammation, hypertension, and cardiovascular disease [6,7,8,9].

We have reviewed the literature on CML detection from 1999 to 2016 [10], and dietary CML is especially prevalent in dairy products. Many studies reported similar conclusions. For example, Hull et al. reported that heat-treated dairy products have higher CML concentrations than other food categories (0–1015 mg/kg protein) [11]. The heating, sterilization, and spray-drying methodologies that are often used in dairy processing can also promote the Maillard reaction. CML is formed in dairy processing by reactive dicarbonyl compounds, such as methyl-glyoxal or glyoxal, which are metabolites/autoxidation products of ascorbate, glucose, Amadori intermediates, polyunsaturated lipids, or Schiff bases with protein [12]. The reactive dicarbonyl compounds are efficient precursors for CML formation [13], and have been of great interest due to the increasing recognition of their links to a lot of health complications by the induction of protein damage [14].

The control of CML levels in food thermal processing using functional ingredients not only promotes the formation of compounds with the desired color and flavor but also reduces the content of off-flavors and potential toxic molecules. Many studies have reported the trapping of scavenging reactive dicarbonyl compounds with some antioxidant compounds, such as polyphenols, flavonoids, and phenolic acids, with beneficial results for the inhibition of CML formation in food thermal processing [15,16]. These phenolic compounds are natural antioxidants in foods and have attracted considerable attention due to their possible in vivo antioxidant activity, which beneficially affect human health. For example, Yang et al. used (+)-catechin to efficiently inhibit methylglyoxal (MGO)-induced Schiff-based CML formation in vitro [17]. We used five phenolic acid compounds to inhibit CML formation and found that the optimal inhibitory effects of caffeic, syringic, ferulic acids, *p*-coumaric, and sinapic acids were 55.5%, 56.5%, 43.3%, 41.1%, and 44.0%, respectively [18]. 

Vinasse is a byproduct from winemaking based on sorghum, wheat, or rice that is not used further in wine production but instead is used as an important source of feed for pigs, cattle, poultry, and others [19]. Chinese baijiu is flavorful and contains potential functional components that are beneficial to humans, especially phenolic compounds. Que et al. reported ten kinds of phenolic compounds in Chinese rice wine as detected by HPLC, including caffeic acid, syringic acid, rutin, (−)-epicatechin, (+)-catechin, gallic acid, vanillic acid, *p*-coumaric acid, ferulic acid, and quercetin [20]. Phenolic acids are correlated with high antioxidant levels and activity, reducing capacity and free radical scavenging activity. However, there is little information regarding the functional components and potential applications in alleviating or preventing the chronic diseases induced by vinasse, which may be related to its disposal in baijiu production. 

In this study, we investigated the inhibitory effects of baijiu vinasse extract and its phenolic acid compounds on the CML formation in dairy food. The extraction conditions of baijiu vinasse were optimized by different extraction solvents, volumes, temperatures, time, and replicates to identify and detect phenolic acid compounds. Based on the structural differences of the phenolic acid compounds, we compared their inhibitory effects with aminoguanidine, which is an effective CML inhibitor in both in vitro and in vivo studies that is highly toxic to diabetic patients [21]. The inhibitory mechanism of the phenolic acid compounds in the model of dairy food was discovered to be the trapping and scavenging of glyoxal. Furthermore, the antioxidant activities of vinasse extract were evaluated by DPPH and ferric ion reducing antioxidant power (FRAP) assays. 

## 2. Results and Discussion

### 2.1. Effects of Different Extraction Conditions on the Total Phenolic Compounds Isolated from Baijiu Vinasse

The concentrations of total phenolic compounds in baijiu vinasse extracts were detected by the Folin–Ciocalteu method to evaluate the different extraction conditions. The extraction solvents were optimized by acetone and double distilled water (DDW), as shown in Figure 1A. The total phenolic compounds were higher in the vinasse extracts from acetone than from DDW and the highest level, at 665.78 mg gallic acid equivalent (GAE)/100 g vinasse extract, was found in the 70% acetone extraction. Because the acetone-water system effectively degraded the protein-polyphenol complex, the extraction rate of phenolic compounds was relatively better using acetone-water than that of other solvents, especially in the matrix containing protein [18]. The levels of total phenolic compounds were 520.77, 561.51, 630.76, 652.92, and 643.63 mg GAE/100 g in the extracts made from 40%, 50%, 60%, 80%, and 90% acetone, respectively. Therefore, with regard to the effectiveness of phenolic compound isolation, 70% acetone was found to be the most effective.

Figure 1B shows the level of total phenolic compounds in the vinasse extracts that used 2 to 10 mL extraction volumes and indicates that the 6 mL 70% acetone extraction was the optimal volume for the extraction of total phenolic compounds. This condition resulted in 544.36 mg GAE/100 g, and the total phenolic compound concentrations were 363.64, 524.36, 530.79, and 508.64 mg GAE/100 g when 2, 4, 8, and 10 mL 70% acetone, respectively, was used.

The effect of extraction temperature was determined using 6 mL 70% acetone in the range of 30 °C to 60 °C. The level of total phenolic compounds was the highest when 6 mL of 70% acetone was used at an extraction temperature of 60 °C, as shown in Figure 1C. The total phenolic compound concentrations were 562.93, 563.65, 573.64, 628.66, 573.63, and 479.36 mg GAE/100 g at the extraction temperatures of 30, 40, 50, 60, 70, and 80 °C, respectively. Interestingly, at temperatures above 60 °C, the concentration of total phenolic compounds decreased gradually. This effect may be due to the decomposition of phenolic compounds by the high temperatures.

The length of extraction was determined with 6 mL 70% acetone at 60 °C from 10 to 60 min, as shown in Figure 1D. The levels of total phenolic compound were 564.36, 565.79, 591.51, 651.53, 603.64, and 547.20 mg GAE/100 g for 10, 20, 30, 40, 50, and 60 min, respectively. The optimal extraction time was clearly 40 min.

The number of extraction replicates was optimized with 6 mL 70% acetone extraction at 60 °C for 40 min repeated 1 to 6 times, as shown in Figure 1E. The supernatant was collected by centrifugal separation, the residue underwent the previous extraction protocol, and the supernatant was collected again. The optimal number of extractions was four, with the level of total phenolic compounds determined to be 674.36 mg GAE/100 g. The concentrations of phenolic compounds were 529.36, 597.22, 600.79, 614.36, and 581.51 mg GAE/100 g following one, two, three, five, and six repetitions of the extraction protocol, respectively. Therefore, the optimal extraction conditions for baijiu vinasse were established as 6 mL 70% acetone heated to 60 °C for 40 min repeated four times.

### 2.2. Evaluation of the Antioxidant Activity of the Vinasse Extract

The antioxidant activities of the vinasse extract were evaluated by the traditional assays, DPPH and FRAP, as shown in Figure 2A. The radical scavenging activity against DPPH radicals was calculated to be a potent 64.37%, indicating strong antioxidant activity. We also used the FRAP assay to evaluate the total antioxidant capacity. The antioxidant capacity of the 70% acetone vinasse extract was 4.63 mM FE(II)/g vinasse extract. The vinasse extract had a good antioxidant capacity, because it included more polyphenolic compounds.

### 2.3. Inhibitory Effect of Baijiu Vinasse Extract on CML Formation

The inhibitory effect of the optimal vinasse extract was investigated. Figure 2B shows that the vinasse extract displayed a marked 43.2% inhibitory effect on CML formation. We compared the inhibitory effects of the vinasse extract to that of aminoguanidine, which was an effective CML inhibitor, in both in vitro and in vivo studies. The inhibitory rate of aminoguanidine on CML was 44.3%, which was not significantly different from the vinasse extract. We have compared CML inhibition with the previous papers, and summarized literature on CML inhibition by different substances in Table 1. Compared to different substances used for CML inhibition, the baijiu vinasse extract in the study exhibited an acceptable inhibitory effect. The baijiu vinasse extract had many kinds of phenolic compounds, but the total content of phenolic compounds was low compared with other extractions. Further, the baijiu vinasse extract had a variety of phenolic compounds that could remove or trap dicarbonyl compounds, which could prevent the formation of CML.

### 2.4. Identification of Major Phenolic Acids from the Vinasse Extract

Chinese baijiu has many phenolic compounds, which can be affected by the fermentation, distillation, aging, blending, raw materials, and environmental conditions used during production [19]. In this study, seven major phenolic acids in baijiu vinasse were detected by HPLC-MS/MS. The HPLC of the phenolic acids from the vinasse extract are shown in Figure 3. Peaks 1–3 (retention time (RT) 27.534–32.147 min) with *m*/*z* 387 were dimmers of ferulic acid. Peak 4 (RT 43.497 min) had the molecular ion m/z 168.15 [M + H]^+^, peak 5 (RT 44.609 min) had the molecular ion m/z 354.31 [M + H]^+^, and peak 6 (RT 48.732 min), suggesting that these peaks were vanillic acid, chlorogenic acid, and syringic acid, respectively. Peaks 7–9 had the molecular ion *m*/*z* 224.21, 164.16, and 180.15 [M + H]^+^, with the RT 51.608, 53.845, and 55.887 min, suggesting that these peaks were sinapic acid, *p*-coumaric acid, and caffeic acid, respectively. Peak 13 [M + H]^+^ (RT 64.477 min) had the molecular ion *m*/*z* 194.18 [M + H]^+^, suggesting that this peak was ferulic acid. Peaks 10–12 (RT 55.345–59.324min) with a *m*/*z* 245 were syringic acid +HC**OO**H. Peaks 14–15 (RT 73.564–74.973 min) with a *m*/*z* 343 were 8’-5’-DiFA.

The concentrations of the major phenolic acid compounds were determined by quantitative analysis with a standard curve. The vanillic acid had the highest concentration at 15.137 mg/kg as measured by HPLC analysis. The concentrations of chlorogenic, sinapic, *p*-coumaric, caffeic, ferulic, and syringic acid were 1.747, 2.166, 0.973, 6.361, 1.674, and 0.543 mg/kg, respectively, as shown in Table 2. Phenolic compounds are well known to have antioxidant and free radical scavenging effects. The results suggested that there were abundant phenolic acids in the vinasse extract, which results in strong antioxidant capacity and free radical scavenging capacity. This result was consistent with the DPPH and FRAP results.

### 2.5. The Trapping of Glyoxal (GO) by the Major Phenolic Acids Extracted from Vinasse

GO has been indicated to be an important intermediate product of CML that is formed by glucose oxidation. Phenolic acid compounds can trap GO efficiently in the model of dairy food. An additive reaction was used to quantify mono- and di-GO conjugated phenolic acid compounds in the samples, which are the major mechanisms preventing CML formation. The determination of the adduct formation of the major phenolic acid compounds in the phenolic acid−GO system was performed by HPLC-MS/MS. The extracted ion chromatography (EIC) and mass spectrometry (MS) analyses of samples for the phenolic acids reacted with GO are shown in Figure 4. The vanillic, chlorogenic, caffeic, and ferulic acid could react with GO at the trapping efficiencies of up to 47.74%, 43.65%, 21.17%, and 49.91%, respectively. Furthermore, the adduct products are shown in Figure 5. However, the adduct products of syringic acid, sinapic acid, and *p*-coumaric acid with GO are not found in Figure 4 because they might not react with GO when heated to 120 °C for 30 min.

In order to further explore the inhibition mechanism, the remaining GO was derived from 1,2-diaminobenzene to 1,4-quinoxaline, and the trapping rate of GO by phenolic acid was calculated mainly by the amount of remaining GO [29]. The HPLC chromatograms of syringic acid, sinapic acid, and *p*-coumaric acid reacted with GO were studied. More than 50% of GO was trapped within 30 min by syringic acid, sinapic acid, and *p*-coumaric acid, and the scavenging efficiency could be up to 55.63%, 63.17%, and 65.76%, respectively. The GO could be trapped by phenolic acid compounds and was the major mechanism to prevent CML formation. The inhibition of CML formation might involve different mechanisms, including reactive carbonyl trapping [30], antioxidant activity [31], sugar autoxidation inhibition [32], and amino group binding inhibition/competition [33,34].

## 3. Materials and Methods

### 3.1. Materials

Analytical standards *N*^ε^-carboxymethyl-lysine (CML) and glyoxal (GO) were obtained from J&K Scientific Co., Ltd. (Beijing, China), and Yuan Ye Biotechnology Co., Ltd. (Shanghai, China), respectively. The model of dairy food was prepared by sodium caseinate and d-glucose, which were purchased from West Dragon Science Co., Ltd. (Shantou, Guangdong, China). Standards of *p*-coumaric, ferulic, caffeic, syringic, sinapic, vanillic, and chlorogenic acids (analytical grade, ≥98%) were purchased from Shanghai Tian Biotechnology Co., Ltd. (Shanghai, China). The organic reagents like acetone, methanol, and acetonitrile were purchased from Sinopharm Chemical Reagent Co., Ltd. (Beijing, China). The other reagents of gallic acid, formic acid, sodium carbonate, disodium hydrogen phosphate dodecahydrate and sodium dihydrogen phosphate dehydrate, sodium borate, sodium hydroxide, and sodium borohydride were purchased from J&K Scientific, Co., Ltd. (Beijing, China). Folin–Ciocalteu reagent (analytical grade), 1,1-diphenyl-2-picrylhydrazyl (DPPH, analytical grade), and *O*-phenylenediamine was purchased from Source Biological Technology Co., Ltd. (Shanghai, China). DDW was prepared using a WaterPro water purification system (Labconco Corp., Kansas City, MO, U.S.A.). The baijiu vinasses were provided by a large local baijiu manufacturer.

### 3.2. Equipment

Samples were centrifuged using a Velocity 18R high-speed refrigerated centrifuge (Hitachi Koki Co., Ltd., Tokyo, Japan). CML was analyzed using 1260 infinity high-performance liquid chromatography-tandem mass spectrometry (HPLC/MS-MS, Agilent Technologies, Inc., Santa Clara, CA, USA). The Inertsil ODS C_18_ column (150 × 4.6 mm, 4.6 μm) was purchased from Thermo Fisher Scientific Co., Ltd. (Shanghai, China). Samples were dried using an R-210 rotary evaporator (Büchi Labortechnik AG, Flawil, Switzerland). Ultraviolet–visible (UV–vis) spectra over 200–800 nm were recorded on a Cary50-Bio UV spectrometer (Victoria, Australia). 

### 3.3. Preparation of the Model of Dairy Food

The model system was constructed by the method published by Nguyen et al. with few modifications [35]. The sugar and lysine concentrations were set to approximately 10:1 to simulate bovine milk. Then, d-glucose (27 g) and sodium caseinate (30 g) were added into phosphate buffer (0.1 M, 1L, pH = 7.4), stirring constantly for 8 h at room temperature. The resultant solution was stored overnight at 4 °C. Aliquots of model solution (6 mL) were transferred into tubes for further heating at 120 °C for 30 min.

### 3.4. Optimization of Extraction Conditions for Vinasse

A single factor study was used to determine the optimum extraction conditions of vinasse by identifying or detecting total phenolic contents under various conditions and optimizing the extraction solvent, volume, temperature, and time.


The extraction solvent was especially important in this study, and different concentrations of acetone (40%, 50%, 60%, 70%, 80%, and 90%) were used, with DDW as a control. Typically, 0.20 g of vinasse was accurately weighed and then added to 4 mL of the extraction solvent. After magnetic stirring at 60 °C for 30 min, it was centrifuged at 4000 rpm within 5 min, and the supernatant was collected. The centrifugation was repeated two times, and the extracts were combined and evaporated to near dryness in a vacuum rotary evaporator at 50 °C. The residue was redissolved in 5 mL of extraction solvent for analysis. In addition, 2, 4, 6, 8, and 10 mL of 70% acetone were used as extraction volumes, and 30, 40, 50, 60, 70, and 80 °C were chosen as extraction temperatures with a mixture of vinasse (0.20 g) and 70% acetone (6 mL). Under the optimal conditions, the vinasse mixtures were extracted at 60 °C for 10, 20, 30, 40, 50, or 60 min. Furthermore, the extraction was repeated one to six times and the best condition was chosen. Each vinasse extract was evaluated by determining the total phenolic compound content.

### 3.5. Inhibitions of Vinasse Extract on CML Formation

The inhibitions were determined by adding 1.5 mL of vinasse extract to the equivalent volume of the model of dairy food. To evaluate the inhibitory effect, a control experiment was performed by adding 1.5 mL of 70% acetone without vinasse extract. We used aminoguanidine as a positive control to evaluate the inhibition on CML.

### 3.6. Determination of CML

The CML detection was performed as in the previous research methods of our group by HPLC-MS/MS analysis [18]. The pretreatment of samples was performed according to the methods of Hull et al., with some modifications [11]. A quantity of sample, equivalent to 2 mg protein, was added to sodium borate buffer (0.5 M, pH 9.2, 400 μL) and sodium borohydride (2 M, 200 μL) overnight at 4 °C. The mixture was isolated by chloroform:methanol (2:1, *v*/*v*) and then hydrolyzed with 6 M HCl (1 mL) at 110 °C for 24 h. The mixture was evaporated to near dryness in a vacuum rotary evaporator at 80 °C and redissolved in 1 mL of DDW for HPLC-MS/MS analysis. 

### 3.7. Determination of Total Phenolic Content in Vinasse Extract

The total phenolic compound content in vinasse extract was evaluated by the Folin–Ciocalteu method. Following the method of Alves et al. [36], 100 μL vinasse extract was mixed with 400 μL DDW and 0.25 mL 1 M Folin–Ciocalteu reagent before adding sodium carbonate (7.5 g/100 mL, 1.25 mL). Within incubation for 120 min at room temperature, we measured the ultraviolet absorbance of the mixture by an ultraviolet spectrophotometer with a maximum ultraviolet absorption wavelength of 725 nm.

### 3.8. Determination of Antioxidant Capacity in Vinasse Extract

The free radical scavenging ability was evaluated by the DPPH method. Typically, the vinasse extract (150 μL) was added to 2.85 mL of DPPH (6.6 × 10^−5^ M) in dark conditions, the mixture was incubated for 120 min, and the ultraviolet maximum absorbance was measured at 515 nm. The following equation was calculated for the DPPH inhibitory rate:Inhibition% = 1 − A_s_/A_c_ × 100(1)
where A_s_ and A_c_ are the absorbance value of the sample and the blank, respectively.

The FRAP method of determining total antioxidant capacity is based on the principle that antioxidants can reduce the blue Fe^2+^-TPTZ molecule produced by Fe^3+^-TPTZ under acidic conditions, and then the level of the blue Fe^2+^-TPTZ can be quantified at 593 nm to obtain the total antioxidant capacity of the sample. For the FRAP method, the total antioxidant capacity was expressed as the concentration of the FeSO_4_ standard solution. Its results are expressed as mM Fe(II)/g vinasse extract.

### 3.9. Determination of the Trapping/Scavenging of Phenolic Acids for GO by HPLC-MS/MS

The phenolic acid compounds (5 mM, 0.5 mL), including vanillic, chlorogenic, syringic, sinapic, *p*-coumaric, caffeic, and ferulic acid, were mixed with GO (5 mM, 0.5 mL) before the mixture was incubated for 30 min at 120 °C. After the reaction, the mixture was cooled quickly. Then, 0.25 mL 1,2-diaminobenzene (100 mM) and 0.25 mL 2,3-dimethylquinoxaline (5 mM, internal standard) were added to the phenolic acid-GO mixture, and the reaction continued for 30 min at room temperature.

HPLC-MS/MS analysis was performed using a 1260 Infinity (Agilent) and the Inertsil ODS C_18_ column. For the isometric elution mode, the mobile phases A:B (1:1) were DDW:methanol, and the flow rate was 0.6 mL/min. The mass spectra were used in a full-scan mode with ion monitoring *m*/*z* set to 100 to 1000, capillary voltage set to 4000 V, and ionization source set to 300 °C. The ultraviolet detection wavelength was at 315 nm.

### 3.10. Identification of Major Phenolic Acids by HPLC-MS/MS

The phenolic composition was determined using HPLC−MS/MS [37]. The phenolic compounds were separated by an Inertsil ODS C_18_ column and 1260 Infinity HPLC-MS/MS (Agilent). For binary gradient elution, the mobile phases A and B were 0.1% formic acid in 99.9% DDW and 100% methanol, respectively. The mobile phase began with 90% A from 0 to 110 min, followed by increases in B to 100% at 110 min and 10% B at 140 min. Then, the mobile phase was maintained at 10% B until 150 min. The flow rate was 0.3 mL/min, the sample injection volume was 20 μL, and the ultraviolet Varian detector was set at 320 nm. The mass spectra were acquired under the same conditions as for quantifying GO.

### 3.11. Statistical Analysis

The experimental data are expressed as the mean ± standard deviation and were plotted by Origin 8.5 software (Origin Lab Co., Microcal, MA, USA).

## 4. Conclusions

In conclusion, the baijiu vinasse extract has a strong antioxidant capacity and a clear inhibition on CML formation in dairy food. The antioxidant activities of vinasse extract was evaluated by the FRAP and DPPH methods. These assays demonstrated that the 70% acetone vinasse extract had the potent radical scavenging activity against DPPH radicals of 64.37% and the antioxidant capacity of 4.63 mM of FE(II)/g. The extraction conditions of baijiu vinasse were optimized based on total phenolic acid concentrations. Similar to the aminoguanidine positive control, the inhibitory effect of the optimally extracted vinasse extract could reach 43.2%. The HPLC-MS/MS profiles of the vinasse extract indicated that vanillic, chlorogenic, sinapic, *p*-coumaric, caffeic, ferulic, and syringic acid were the seven major phenolic acid compounds. Furthermore, the inhibitory mechanism of phenolic acid compounds in the model of dairy food was determined to be the trapping of glyoxal. The seven phenolic acid compounds could form the adducts of mono- and di-GO conjugated phenolic acid compounds, and the average trapping/scavenging rate was strong at 49.57%. The effective utilization of vinasse extracts will not only reduce environmental pollution but also comprehensively and effectively use resources to reduce waste. Vinasse extract could be used to inhibit the formation of other food-borne advanced glycation end products and to reduce the risk of developing chronic diseases.

## Figures and Tables

**Figure 1 molecules-24-01526-f001:**
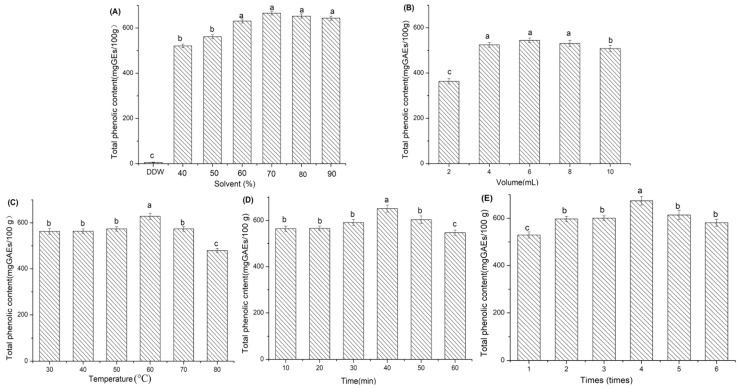
Optimization of extraction conditions based on the concentration of total phenolic compounds. The effects of (**A**) solvent, (**B**) volume, (**C**) temperature, (**D**) time, and (**E**) replicates are shown (values are presented in mean ± standard deviation (*n* = 3)). Different superscript letters (a–c) in the coordinate axis denotes significant differences at *p* < 0.001 (* *p* < 0.05).

**Figure 2 molecules-24-01526-f002:**
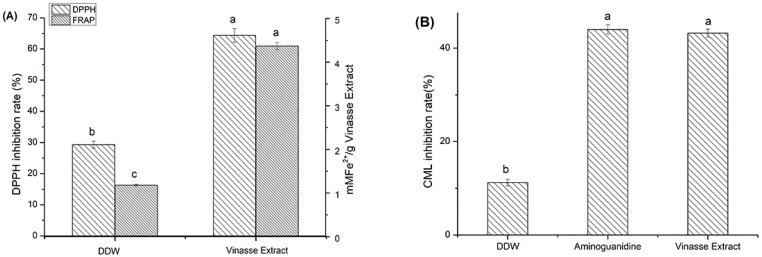
(**A**) The antioxidant capacities of the vinasse extract as measured by DPPH and ferric ion reducing antioxidant power (FRAP). (**B**) The inhibitory rate of CML formation by vinasse extract, double distilled water (DDW), and aminoguanidine (values are presented in mean ± standard deviation (*n* = 3)). Different superscript letters (a–c) in the coordinate axis denotes significant differences at *p* < 0.001 (* *p* < 0.05).

**Figure 3 molecules-24-01526-f003:**
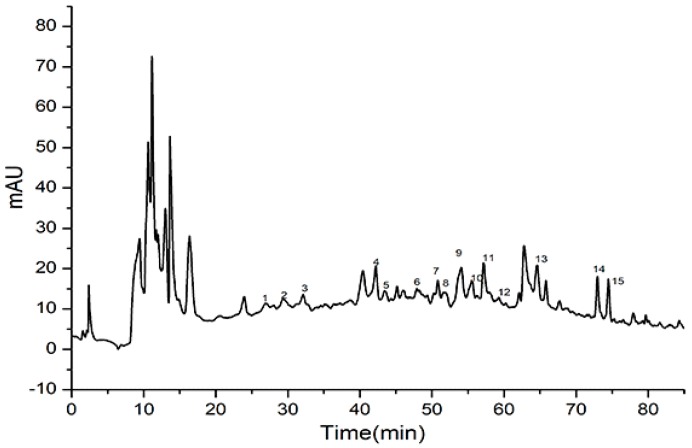
HPLC chromatograms of the major phenolic acid compounds in the vinasse extract. Peaks 1–3 are dimer ferulic acid. Peaks 4–6 are vanillic acid, chlorogenic acid, and syringic acid, respectively. Peaks 7–9 are sinapic acid, *p*-coumaric acid, and caffeic acid, respectively. Peaks 10–12 are syringic acid + HC**OO**H. Peak 13 is ferulic acid. Peaks 14–15 are 8’-5’-DiFA.

**Figure 4 molecules-24-01526-f004:**
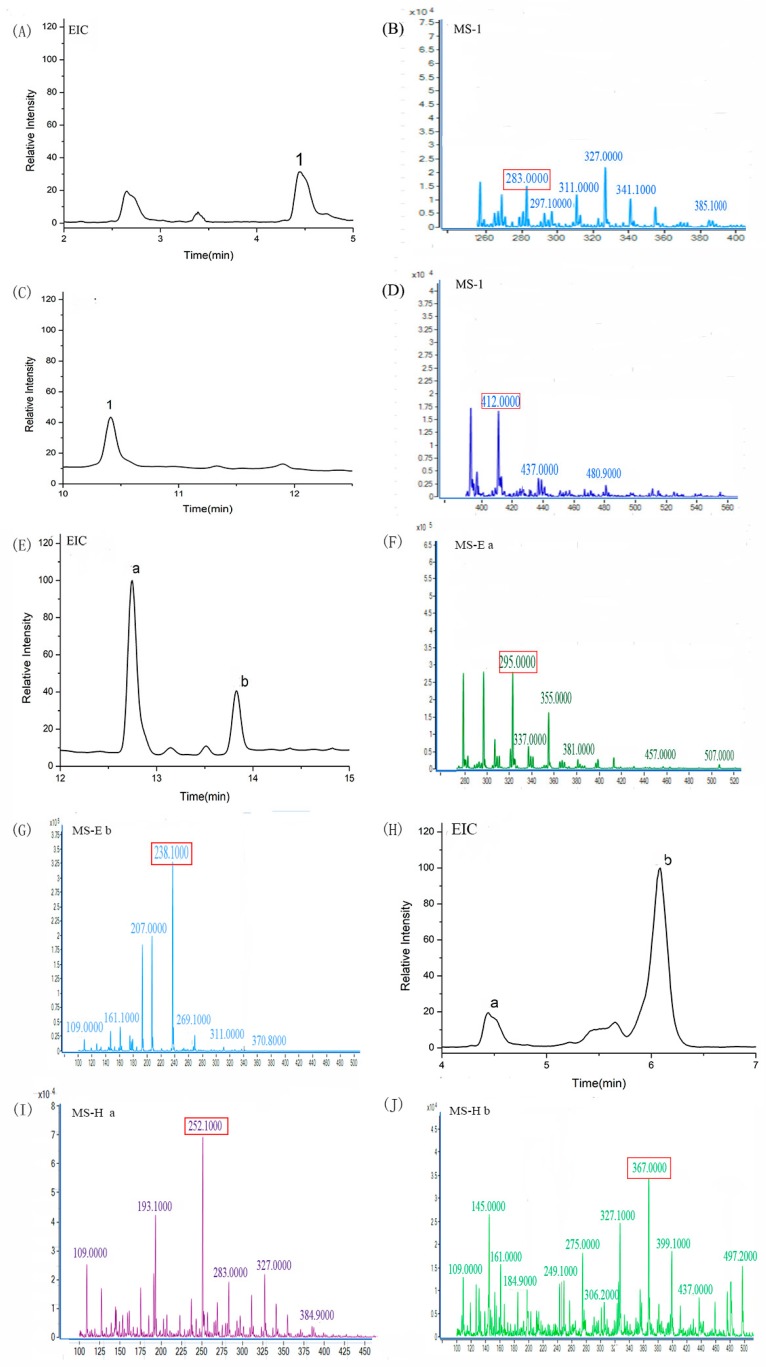
The extracted ion chromatography (EIC) and mass spectrometry (MS) analyses of samples for the phenolic acids reacted with glyoxal (GO). (**A**) EIC for vanillic acid reacted with GO, (**B**) MS for vanillic acid reacted with GO, (**C**) EIC for chlorogenic acid reacted with GO, (**D**) MS for chlorogenic acid reacted with GO, (**E**) EIC for caffeic acid reacted with GO, (F/G) MS for caffeic acid reacted with GO, (**H**) EIC for ferulic acid reacted with GO, (**I**/**J**) MS for ferulic acid reacted with GO.

**Figure 5 molecules-24-01526-f005:**
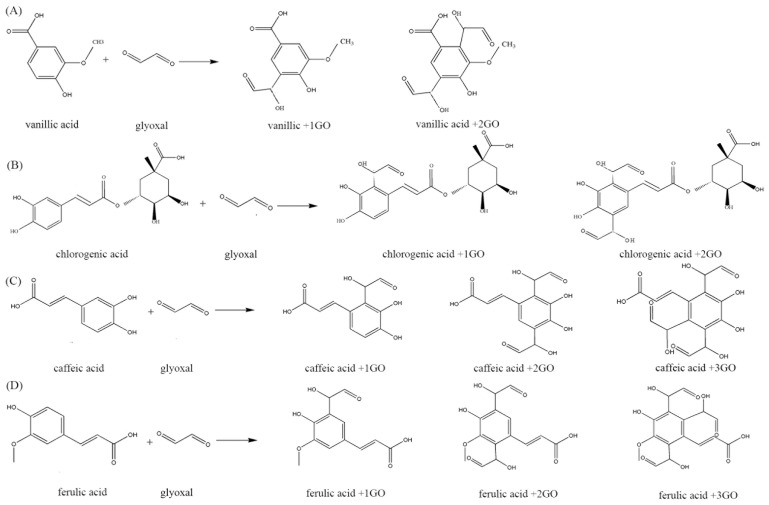
The adduct products of (**A**) vanillic, (**B**) chlorogenic, (**C**) caffeic, and (**D**) ferulic acids following reaction with GO.

**Table 1 molecules-24-01526-t001:** Summary of literature reports on CML inhibition by different substances.

Samples	Extraction	Main Components	Inhibition Rate	Reference
Highland barley bran	Highland barley bran extract	Ferulic, syringic, sinapic, *p*-coumaric, and caffeic acids	45.58%	[18]
Sugarcane molasses	Sugarcane molasses extract	Catechin, vanillic acid, syringic acid, tricin 7-*O*-glucoside, and *p*-coumaric acid	34.8–85.8%	[22]
Chrysanthemum	Chrysanthemum species extracts	Chlorogenic acid, flavonoid glucoside varieties, apigenin, caffeic acid, luteolin, and kaempferol	28.5–79.1%	[23]
20 microalgae	Microalgal extracts	Carotenoids in chlorella and unsaturated fatty acids, mainly of linoleic acid, arachidonic acid, and eicosapentaenoic acid	81.76–91.68%	[24]
Nine dried edible or medicinal flowers	Several floral herbal infusions	Phenolic compounds	9.5–96.4%	[25]
Distilled residues of rice spirit, sweet potato spirit, and barley spirit	Fermentation byproducts	Lysine and arginine protein, phenolic, total solids, and total volatile solids determinations	31.1–96.7%	[26]
Green pepper, apricot, hazelnut, peach, sour cherry, sesame, almond, and pomegranate	Fruit and vegetable seed extracts	Phenolic acids, benzoic acids, flavonoids, such as p-hydroxybenzoic acid, syringic acid, vanillic acid, *p*-coumaric acid, caffeic acid, ferulic acid, protocatechuic acid, gallic acid, gentisic acid, sinapinic acid, and ellagic acid	20–78.6%	[27]
Mung bean, black bean, soybean, and cowpea	Mung bean extract	Constituents vitexin and isovitexin	65.3–81.2%	[28]

**Table 2 molecules-24-01526-t002:** The concentrations of the major phenolic acid compounds in the vinasse extract.

Phenolic Acid Compounds	Retention Time (min)	Content (mg/kg)
vanillic acid	43.497	15.137 ± 0.036
chlorogenic acid	44.609	1.747 ± 0.012
syringic acid	48.732	0.543 ± 0.003
sinapic acid	51.608	2.166 ± 0.002
*p*-coumaric acid	53.845	0.973 ± 0.001
caffeic acid	55.887	6.361 ± 0.014
ferulic acid	64.477	1.674 ± 0.011

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
