# Peer review of "Baijiu Vinasse Extract Scavenges Glyoxal and Inhibits the Formation of *N*^ε^-Carboxymethyllysine in Dairy Food"

_molecules, 2019, doi:10.3390/molecules24081526_

Round 1
Reviewer 1 Report
The present manuscript evaluates the ability of a baijiu vinasse extract to scavenge glyoxal and inhibit the formation of an AGE (N-carboxymethylisine). The ability of different phenolic extracts, as well as pure phenolic compounds, to inhibit AGE formation and to scavenge AGEs has been already demonstrated (also in vivo). In this line, the novelty of the work could be considered as limited. However, the major concern arises from the fact that a deeper discussion of the results is missed throughout the manuscript. In this regard, the authors could compare their results with previous and similar papers. Another major concern arises from the fact that a more detailed characterization of the extract would be appreciated; which are the peaks between 10-20 min in the chromatogram of the extract that indeed seem to be more abundant than the components identified by the authors? Please consider that to claim that the activity to scavenge and inhibit AGE formation are due to phenolics, pure compounds must be tested, and at this point the properties observed are related to the whole extract. Therefore, these analyses must be done or the title changed.
Please, include “n” (number of samples and replicates of each sample in all figure legends).
Author Response
Response to Reviewers
Reviewers' comments:
Reviewer 1
1. The major concern arises from the fact that a deeper discussion of the results is missed throughout the manuscript. In this regard, the authors could compare their results with previous and similar papers. Another major concern arises from the fact that a more detailed characterization of the extract would be appreciated; which are the peaks between 10-20 min in the chromatogram of the extract that indeed seem to be more abundant than the components identified by the authors? Please consider that to claim that the activity to scavenge and inhibit AGE formation are due to phenolics, pure compounds must be tested, and at this point the properties observed are related to the whole extract. Therefore, these analyses must be done or the title changed.
Answer: We have compared CML inhibition with the previous papers, and summarized literature on CML inhibition by different substances in Table 1. Compared to different substances used for CML inhibition, the Baijiu vinasse extract in the study exhibited a acceptable inhibitory effect. The baijiu vinasse extract had a variety of phenolic compounds that could remove or trap dicarbonyl compounds, which could prevent the formation of CML. We have added it on line 150-153 of the manuscript.
Table 1 Summary of literature reports on CML inhibition by different substances.
Samples | Extraction | Main Components | Inhibition Rate | Reference |
Highland barley bran | Highland barley bran extract | Ferulic, syringic, sinapic, p-coumaric, and caffeic acids | 45.58% | [1] |
Sugarcane molasses | Sugarcane molasses extract | Catechin, Vanillic acid, Syringic acid, Tricin 7-O-glucoside, p-Coumaric acid | 34.8%-85.8% | [2] |
Chrysanthemum | Chrysanthemum species extracts | Chlorogenic acid, flavonoid glucoside varieties, and apigenincaffeic acid, luteolin, and kaempferol | 28.5%-79.1% | [3] |
20 microalgae | Microalgal extracts | Carotenoids in Chlorella and unsaturated fatty acids, mainly of linoleic acid, arachidonic acid and eicosapentaenoic acid | 81.76–91.68% | [4] |
Nine dried edible or medicinal flowers | Several floral herbal infusions | Phenolic compounds | 9.5%-96.4% | [5] |
Distilled residues of rice spirit, sweet potato spirit, and barley spirit | Fermentation byproducts | Lysine and arginine Protein, phenolic, total solids, total volatile solids determinations | 31.1%-96.7% | [6] |
Green pepper, apricot, hazelnut, peach, sour cherry, sesame, almond and pomegranate | Fruit and vegetable seed extracts | Phenolic acids, benzoic acids, flavonoids,such as p-hydroxybenzoic acid, syringic acid; vanillic acid; p-coumaric acid; caffeic acid; ferulic acid; protocatechuic acid; gallic acid; gentisic acid; sinapinic acid; ellagic acid | 20-78.6% | [7] |
Mung bean, black bean, soybean and cowpea | Mung bean extract | Constituents vitexinand isovitexin | 65.3%-81.2% | [8] |
Baijiu vinasse | Vinasse extract | Vanillic, chlorogenic, p-coumaric, sinapic, caffeic, ferulic, syringic acids | 43.2% | This Article |
References
1. Liu, H.; Chen, X.; Zhang, D.; Wang, J.; Wang, S.; Sun, B., Effects of highland barley bran extract rich in phenolic acids on the formation of N(epsilon)-carboxymethyllysine in a biscuit model. Journal of agricultural and food chemistry 2018, 66, (8), 1916-1922.
2. Yu, P.; Xu, X. B.; Yu, S. J., Inhibitory effect of sugarcane molasses extract on the formation of N(epsilon)-(carboxymethyl)lysine and N(epsilon)-(carboxyethyl)lysine. Food Chem 2017, 221, 1145-1150.
3. Tsuji-Naito, K.; Saeki, H.; Hamano, M., Inhibitory effects of Chrysanthemum species extracts on formation of advanced glycation end products. Food Chemistry 2009, 116, (4), 854-859.
4. Sun, Z.; Peng, X.; Liu, J.; Fan, K.-W.; Wang, M.; Chen, F., Inhibitory effects of microalgal extracts on the formation of advanced glycation endproducts (AGEs). Food Chemistry 2010, 120, (1), 261-267.
5. Ho, S.-C.; Chang, P.-W.; Tong, H.-T.; Yu, P.-Y., Inhibition of fluorescent advanced glycation end-products and N-carboxymethyllysine formation by several floral herbal infusions. International Journal of Food Properties 2013, 17, (3), 617-628.
6. Ye, X.-J.; Ng, T. B.; Nagai, R., Inhibitory effect of fermentation byproducts on formation of advanced glycation end-products. Food Chemistry 2010, 121, (4), 1039-1045.
7. Mesias, M.; Navarro, M.; Gokmen, V.; Morales, F. J., Antiglycative effect of fruit and vegetable seed extracts: inhibition of AGE formation and carbonyl-trapping abilities. Journal of the science of food and agriculture 2013, 93, (8), 2037-44.
8. Peng, X.; Zheng, Z.; Cheng, K.-W.; Shan, F.; Ren, G.-X.; Chen, F.; Wang, M., Inhibitory effect of mung bean extract and its constituents vitexin and isovitexin on the formation of advanced glycation endproducts. Food Chemistry 2008, 106, (2), 475-481.
We tried our best to optimize the separation conditions by HPLC-MS/MS. But the peaks were still obvious between 10 and 20 min. The peaks between 10-20 min in the chromatogram of the extract may be solvent and impurity peaks, because of the phenolic acids were very low in the extract. In addition, we did not use the standards of phenolic acids to inhibit CML in the study, so we changed the title to “Baijiu vinasse extract scavenge glyoxal and inhibit the formation of Nε-carboxymethyllysine in dairy food”. Thank you very much.
2. Please, include “n” (number of samples and replicates of each sample in all figure legends).
Answer: We have added it in the figure legends. Thank you very much.

Reviewer 2 Report
Nowadays, the idea of converting agro-industrial waste into energy and other useful chemicals represents a research area with great potential and opportunities, but the technologies used for their exploitation must be environmental friendly and sustainable from the economically point of view. Although several agricultural residues can be safely discarded in the environment, the disposal of vinasse produce a lot of inconveniences for the ecosystem, due to its significant nutritional value and high concentration of organic compounds that confers a high biochemical oxygen demand for the waste degradation.
The inhibitory effects of baijiu vinasse extract and its phenolic acid compounds on the Nε-carboxymethyllysine (CML) formation from dairy food were investigated. Furthermore, the inhibitory mechanism of the phenolic acid compounds in the model of dairy food was discussed by the trapping and scavenging of glyoxal.
The manuscript addresses an interesting topic and prior to publication only some minor corrections are required.
The article is generally well written, the English language is appropriate and understandable, only minor spell check being required. Also, the methods and reagents are described with sufficient details to allow another researcher to reproduce the results.
It would be useful to include a short list of abbreviations.
Please provide a better-quality version for Figure 5
Why did you choose acetone: water mixture for the extractions?
Could the process be scaled up at industrial level? Is the process economically feasible?
Line 65: please correct the word “regarding”
Line 135: I suggest to reformulate these phrases to avoid the expression “extraordinary effort”.
Author Response
Response to Reviewers
Reviewers' comments:
Reviewer 2
1. It would be useful to include a short list of abbreviations.
Answer: We have added a short list of abbreviations in Table 1 of the revised manuscript. Thank you very much.
2. Please provide a better-quality version for Figure 5
Answer: We have revised Fig.5. Thank you very much.
3. Why did you choose acetone: water mixture for the extractions?
Answer: Because acetone-water system effectively degraded protein-polyphenol complex, the extraction rate of phenolic compounds was relatively better using acetone-water than that of other solvent, especially in the matrix containing protein. We have added it on line 100-103 of the revised manuscript. Thank you very much.
4. Could the process be scaled up at industrial level? Is the process economically feasible?
Answer: This process can scale up at industrial level. Chinese Baijiu is flavorful and contains potential functional components that are beneficial to humans, especially phenolic compounds. Baijiu is an important part of the Chinese food industry, but its vinasse is a byproduct from winemaking based on sorghum, wheat or rice that is not used further in wine production but instead is used as an important source of feed for pigs, cattle, poultry, and others. It has a very good development prospect and direction in the processing and production of food enterprises and the health products industry, providing a theoretical basis for the secondary comprehensive utilization of vinasse. Thank you very much.
5. Line 65: please correct the word “regarding”
Answer: We have revised it on line 79 of the revised manuscript. Thank you very much.
6. Line 135: I suggest to reformulate these phrases to avoid the expression “extraordinary effort”.
Answer: We have changed it to “In this study, seven major phenolic acids in baijiu vinasse were detected by HPLC-MS/MS” on line 159-161 of the revised manuscript. Thank you very much.

Round 2
Reviewer 1 Report
The revised present manuscript has been improved.
Please, in the table 2 consider to delete the reference to the present work, and add a sentence highlighting the result obtained by the authors in comparison with previous works.
Author Response
Response to Reviewers
Reviewers' comments:
Please, in the table 2 consider to delete the reference to the present work, and add a sentence highlighting the result obtained by the authors in comparison with previous works.
Answer: We have deleted the reference, and added a sentence highlighting the result in comparison with previous works. We have added it on line 138-140 of the manuscript. Thank you very much.